# Lifestyle, Quality of Life, and Health Promotion Needs in Mexican University Students: Important Differences by Sex and Academic Discipline

**DOI:** 10.3390/ijerph17218024

**Published:** 2020-10-31

**Authors:** Georgina Mayela Núñez-Rocha, Cynthia Karyna López-Botello, Ana María Salinas-Martínez, Hiram V. Arroyo-Acevedo, Rebeca Thelma Martínez-Villarreal, María Natividad Ávila-Ortiz

**Affiliations:** 1Facultad de Salud Pública y Nutrición, Universidad Autonoma de Nuevo Leon., Nuevo León 66455, Mexico; cynthia_k12@hotmail.com (C.K.L.-B.); amsalinasmartinez@gmail.com (A.M.S.-M.); natividad.avilao@uanl.mx (M.N.Á.-O.); 2Unidad de Investigación Epidemiológica y en Servicios de Salud, Instituto Mexicano del Seguro Social., Nuevo León 64360, Mexico; 3Escuela de Salud Pública, Recinto de Ciencias Médicas, Universidad de Puerto Rico, San Juan, PR 00931, USA; hiram.arroyo1@upr.edu; 4Centro Universitario de Salud, Universidad Autonoma de Nuevo Leon, Nuevo León 66455, Mexico; rebeca.martinezv@uanl.mx

**Keywords:** lifestyle, quality of life, health promotion, university students, Mexico

## Abstract

Few studies have evaluated and contrasted the lifestyles and quality of life of university students by academic discipline. We compared university students’ lifestyle and quality of life, and schools’ compliance with health promotion guidelines. Then, needs were ranked and prioritized. This was a cross-sectional study carried out in a public university in Northeastern Mexico. Higher education students with no visual or hearing impairment from six different academic disciplines were included (*N* = 5443). A self-administered and anonymous questionnaire was applied that included the HPLP (Health-Promoting Lifestyle Profile) and SF-12 scales. A check list was employed for measuring 26 on-site schools’ compliance with health promotion guidelines, and needs were ranked using Z-scores. The mean lifestyle was 53.9 ± 14.8 and the mean quality of life was 69.7 ± 5. Men had healthier lifestyles with more exercise and better stress management. The mean compliance with health promotion guidelines was 58.7%. Agricultural Sciences students had the highest need for improving both lifestyle and quality of life. Arts, Education, and Humanities, Engineering and Technology, and Social and Administrative Sciences schools ranked first in need for health promotion actions. The methodology used allowed hierarchization of areas requiring planning and implementation of specific actions, and the results indicated that healthy lifestyles and quality of life should be a priority.

## 1. Introduction

More than 50% of Mexico’s population exceed the World Health Organization recommended intake limits for sugar sweetened beverages and foods high in saturated fat and/or added sugar [1]. Furthermore, less than 40% exercise in their free time, a percentage which has remained stable in the last five years [2]. These unhealthy lifestyles have contributed to three major public health problems in the country: obesity, diabetes, and hypertension. In Mexican college students, the overweight and obesity rate has been reported between 33.4% and 42.4% [3,4,5]. A health-promoting lifestyle is a set of behaviors related to nutrition, exercise, stress management, health responsibility, interpersonal relationships, and spiritual growth. Social and academic pressure on students can cause an undesirable health-promoting lifestyle with consequences on physical and mental health. A health-promoting lifestyle responds to a personal, conscious, and voluntary decision that is shaped by parents during the first years of life. Later, the decision is influenced by friends, teachers, and the school environment. University students may acquire unhealthy eating habits and become addicted to licit or illicit drugs [6,7]. It is also common that they undervalue exercise [8,9,10,11,12,13], health responsibility [8,14,15], and nutritional issues [8,9,10,11]. Statistics on Mexican college students show a light exercise frequency of 39%, a moderate exercise frequency of 44% and a vigorous exercise frequency of 17% [5]. Forty-six-point five percent dedicate most of their free time to the use of information and communication technologies during the week [16], 30% smoke and 44% consume alcohol [3]. Quality of life is another desirable living condition that refers to physical, social, and emotional well-being. Some studies show that university students can have satisfactory quality of life [17,18,19].

Significant lifestyle and quality of life differences by sex can be anticipated because being either a woman or a man impact on decisions in favor of healthy lifestyles. Female students exercise less [20,21,22], have healthier eating habits and drink less alcohol than male students [21,23]. Women also tend to be more stressed, concerned about body image and less satisfied with their current weight [23,24,25,26]. Either men are less susceptible to peer pressure to be thin or they are less likely to admit the pressure [21]. On the other hand, the health sciences curriculum structure is expected to include more health promotion content than others. Therefore, differences between students from health and non-health sciences disciplines can be anticipated [21,22]. Moreover, load and academic exigency differ by career and dissimilarities in quality of life may occur. Some educational programs require more mental work, others more physical work, or both. Some students do no cope effectively to academic demands and experience more stress and sleep problems. Insufficient sleep in turn leads to fatigue, anxiety, emotional instability, mood swings, and eventually lower quality of life [27].

Public health policies must include health-promoting lifestyles. Healthy universities as an approach to health promotion are essential in higher education institutions [28]. Students spend many hours per day in university facilities, and on-site health promotion programs represent an invaluable opportunity to influence positive lifestyles. A literature review revealed that most undergraduate health promotion interventions were primarily focused on a single health issue in England and Latin America [29]. The World Health Organization (WHO) and the Pan American Health Organization (PAHO) highpoint health promotion principles and values for higher education institutions and have released a series of recommendations [30]. Iberic-American universities have been involved with health promotion actions, some examples are creation of healthy environments, support of health promotion networks, staff training, health promotion education and annual meetings [31,32].

We conducted this research because few studies have evaluated both lifestyle and quality of life together [7,33]. Some have made comparisons by academic discipline [9,18,21,34] and only one included the Hispanic population [22]. And, despite WHO/PAHO health promotion principles and values for higher education institutions, not all universities comply [35]. Therefore, the objective of this study was to compare university students’ lifestyle and quality of life, and schools’ compliance with WHO/PAHO health promotion guidelines according to academic discipline. We hypothesized male and health sciences students had healthier lifestyle and better quality of life than female and non-health sciences students. This study serves as a basis on how knowledge on students’ lifestyles and quality of life can be used for ranking and prioritizing the need for improvement by the educational institution. Also, to direct efforts towards schools that most require WHO/PAHO health promotion activities.

## 2. Materials and Methods

### 2.1. Design and Participants

This was a cross-sectional study carried out in one of the largest public universities in Northeastern Mexico from June to November 2018. The study population consisted of higher education students with no visual or hearing impairment who gave their consent to participate in the survey (*N* = 5443). Six academic disciplines were considered: Arts, Education, and Humanities (AEH) (*n* = 448), Agricultural Sciences (ALS) (*n* = 399), Health Sciences (HS) (*n* = 1391), Natural and Exact Sciences (NES) (*n* = 742), Social and Administrative Sciences (SAS) (*n* = 1257), and Engineering and Technology (ET) (*n* = 1206). The sampling was two-stage and not probabilistic. In the first stage, schools were stratified by academic discipline. In the second stage, the students were selected in schools of each stratum. A visit to the school was scheduled from Monday to Friday at different morning and afternoon hours. The sample size per school was planned to be distributed proportionally to the number of students registered per school. The study was approved by the Ethics and Research Committee of the School of Public Health and Nutrition (16-FaSPyN-SA17). All ethical principles for medical research involving human subjects of the Helsinki declaration were respected.

### 2.2. Study Variables

A self-administered and anonymous questionnaire was designed. It included three sections with questions on lifestyle, quality of life, and sociodemographic information.

#### 2.2.1. Lifestyle

The Spanish version of the Health-Promoting Lifestyle Profile (HPLP) validated by Walker et al. in the Hispanic population of the United States was used [36]. It consisted of 48 items organized into six domains: Nutrition (six items with a Cronbach’s α of 0.77), exercise (five items with a Cronbach’s α of 0.81), health responsibility (10 items with a Cronbach’s α of 0.80), stress management (seven items with a Cronbach’s α of 0.71), interpersonal support (seven items with a Cronbach’s α of 0.77), and self-actualization (transcendence) (seven items with a Cronbach’s α of 0.89). The response options were provided on a Likert scale (1 = never, 4 = routinely). All items were written in the positive sense, with no need to reverse responses. The minimum score was 48 and the maximum 192; this score was transformed into a 0–100 scale for easy understanding; the higher the score, the better the lifestyle. The scores were categorized as follows: <33.3 (equivalent to an average response of “never”) = unhealthy lifestyle; >66.6 (equivalent to an average response of “frequently” and “routinely”) = healthy lifestyle; 33.3 to 66.6 (equivalent to an average response of “sometimes”) = moderately healthy lifestyle.

#### 2.2.2. Quality of Life

The Spanish version of the SF-12 validated by Vilagut et al. in Spain was used [37]. It consisted of 12 items with a physical health component (six items with a Cronbach’s α of 0.54) and a mental health component (six items with a Cronbach’s α of 0.72). The physical health component included two items related to physical functioning, two items related to physical role, one item related to bodily pain, and one item related to vitality. The mental health component contained two items related to emotional role, two items related to mental health, and one item related to social functioning, plus a general health question. The answer options varied between two and six levels depending on the question. Four items were presented negatively (bodily pain, general health, vitality, and being calm or peaceful) and it was necessary to invert the answers so that the higher score, the better the quality of life. The minimum score was 12 and the maximum 47, and the scores were transformed into a 0–100 scale. The higher the score, the better the quality of life. The scores were categorized as follows: < 60 poor, 60 to 69 bellow regular, 70 to 79 regular, 80 to 89 good, 90 to 99 very good and 100 excellent.

#### 2.2.3. Sociodemographic

The sociodemographic items comprised age (years), sex (male/female), and current work status (yes/no). Students were also asked for their opinion on the importance of including a health promotion course (1 = not important, 5 = very important) and on the ideal time for taking the course (1 = at the beginning of the career, 2 = in the middle of the career, 3 = at the end of the career).

#### 2.2.4. Procedures

On the day of the school appointment, the classrooms in class were visited for inviting students to participate after explaining the purpose of the study. The first page of the questionnaire corresponded to the written informed consent, which had to be signed before answering the survey; if they did not wish to collaborate, they were reassured that there would be no academic consequences. The questionnaire could be answered in 20 min, but they were given half an hour to be completed. Three research assistants (graduate students) conducted the field work. They received training and supervision by the principal investigator.

#### 2.2.5. Health Promotion

A check list was used for measuring on-site schools’ compliance with the WHO/PAHO health promotion guidelines, consisting of the following 12 indicators: The school development plan includes environmental health policies, the school offers a health promotion course, the school conducts health promotion research, the school has implemented health promotion programs or campaigns, the school has a health prevention or health care module, the school participates in leadership actions and health advocacy, the school provides a health promotion community service, the school displays no smoking signs, the school displays health promotion posters, the school has areas for sports and exercise, the school cafeteria offers a food menu with healthy choices, and the school promotes sports. An advanced and previously trained Master of Science in Public Health student visited all participating schools and collected data by direct observation and by reviewing documents and records (1 = compliant, 0 = not compliant). The minimum score was 0 and the maximum 12, with the resulting amount expressed as a percentage, e.g., the school achieved 7 out of 12 indicators, 7 × 100/12 = 58%.

### 2.3. Statistical Analysis

Frequencies were obtained for the categorical variables and means and standard deviations were obtained for the non-categorical variables. The Student’s *t*-test of independent samples was used to compare lifestyle and quality of life scores between males and females. One-way ANOVA with post-hoc analysis was used to compare lifestyle and quality of life scores between academic disciplines; when the distribution of the variable was not normal, the corresponding non-parametric tests were used. Lifestyle, quality of life, and school compliance with health promotion guideline needs were ranked by academic discipline using Z scores: Z = (X_i_ − $\bar{x}$)/S, where Xi is the observed value of each measure, $\bar{x}$ is the average, and S is the standard deviation.

## 3. Results

The mean age was 19.0 ± 1.9 years, 59.5% were women, and 24.0% worked in addition to studying. In terms of academic year, 20.9% were in their first, 39.9% in their second, 23.6% in their third, and the rest were in their fourth or higher. The mean lifestyle score was 53.9 ± 14.8 (minimum 0, maximum 100; median 53.5); 19.9% of the students had a healthy lifestyle, 72.8% had a moderately healthy lifestyle, and 7.2% had a non-healthy lifestyle. Health responsibility was the domain with the highest percentage of students with a non-healthy lifestyle (51.8%), followed by exercise (38.8%), stress management (30.2%), nutrition (23.9%), interpersonal support (3.5%), and self-actualization (1.6%). The mean quality of life was 69.7 ± 15 (minimum 3, maximum 100; median 71) and the prevalence of good, very good or excellent quality of life was 30.3%. The mental health component was lower than the physical health component (65.7 ± 18.1 vs. 75.5 ± 15.2, respectively; *p* < 0.001).

### 3.1. Lifestyle and Quality of Life Stratified by Sex

Male students had a higher mean healthy lifestyle than female students (54.6 ± 15.2 vs. 53.3 ± 14.6, respectively; *p* < 0.001). Male students also had a higher mean quality of life (72.2 ± 14.7 vs. 68.0 ± 14.9, respectively). Exercise and stress management scores were higher in men, and interpersonal relationship scores higher in women. All quality of life areas except one were higher in men (Table 1).

### 3.2. Lifestyle and Quality of Life Stratified by Academic Discipline

The total lifestyle and quality of life was similar in all disciplines, but the analysis by domain did show differences. Exercise was higher in HS students than in ALS students, and health responsibility was higher in NES students than in HS, SAS, and ET students (*p* < 0.05). On the quality of life side, bodily pain was lower in HS students than in ET and SAS students, and the mental health component was lower in ALS students than in ET, NES, and AEH students (*p* < 0.05) (Table 2).

### 3.3. Schools’ Compliance with Health Promotion Guidelines

Fifty seven percent of the students said a health promotion course was very important and 91.7% considered mid-career as the ideal time for taking the course. The mean compliance with health promotion guidelines was 58.7% (minimum 43.8% and maximum 95.0%). Schools from the HS academic discipline reached the highest achievement, while those from the AEH academic discipline reached the lowest achievement. A food menu with healthy choices was the indicator with the lowest compliance, whereas development plans with environmental health policies and sports promotion were the indicators with the highest compliance (Table 3).

### 3.4. Ranking of Lifestyle, Quality of Life, and Health Promotion Needs

The ALS schools ranked first in need for improving their students’ lifestyle and quality of life (Figure 1). Meanwhile, the AEH, ET, and SAS schools showed the greatest need for improving compliance with health promotion guidelines (Figure 2).

## 4. Discussion

Lifestyle and quality of life recognition is a very fundamental step for the prioritization of problem solving. In this study, a moderately healthy lifestyle was common, as other authors have found [12,13,38,39,40]. Health responsibility, exercise, and stress management were the most affected lifestyle domains—a lack of exercise having already been found in other reports [8,9,11,12,13]. The quality of life was regular, and the mental was more affected than the physical component. The mental health must be addressed, as it can be a reason for school dropouts, alcohol abuse, or other serious consequences [41,42]. A healthy lifestyle and a high quality of life perception of university students facilitate success in accomplishing academic and daily life goals.

Sex has been identified as a health-promoting lifestyle determinant [9,13,43,44]. We hypothesized male had healthier lifestyle and better quality of life than female students. Our results showed men did have healthier lifestyles with more exercise and better stress management, while interpersonal relationships were better in women. These findings are consistent with other literature reports [12,13,15,43]. Men also had better quality of life than women. Excessive stress affects both physical and mental health, in contrast to interpersonal relationships that release stress through expression of emotions. Therefore, they constitute a good way to positively favor quality of life. It is possible that the less quality of life in female students of our study be due to worse stress management despite better interpersonal relationships. Stressful events have a different impact on women than men [43,44]—the former tend to report more stress [45], which in turn influences the quality of life. There is a need to further investigate the specific reasons in future studies. What is undeniable is that the strategies to improve the quality of life should be gender-based.

We also anticipated lifestyle and quality of life differences by students’ academic discipline. Health issues are expected to be addressed in medical, dentistry, nursing, and nutrition schools. Therefore, we specifically expected health sciences students had healthier lifestyle and better quality of life than non-health sciences students, but total lifestyle and quality of life was similar for all academic disciplines. Differences did occur in some domains, students from HS schools exercised more than students from ALS schools. HS students also showed less bodily pain than ET students. Additionally. ALS students had less mental health than those from the ET, NES, and AEH academic disciplines. Pekmezovic et al. [18] reported students of medical sciences to have lower mental health, emotional role, and vitality than students of social sciences and humanities, natural and exact sciences, and technology/engineering. Backhaus et al. [34] showed that students of health sciences and education and languages have the best physical health, and that those from economics and law, engineering, and mathematics/biology disciplines the worst mental health.

The result of health responsibility was striking. Low scores were present in both sexes and in all academic disciplines. CNE students obtained a significantly higher score than students from other three disciplines (HS included), but this was still below 50 on a scale from 0 to 100 points. Two arguments arise. One, the university health promotion system has failed to raise awareness on the students about being health responsible. The other, the promotion of healthy lifestyles involves responsibility on the part of the individual. In other words, it is a matter of co-responsibility [46,47,48]. Particularly, students of the health sciences require special attention because they must lead by the example. University students are young and may feel invulnerable. Many are living away from home for the first time and need to start facing responsibility for their personal health, lifestyle, and behaviors. As youngsters establish identities separate from parents, relationships with peers often become important and interpersonal relations can affect the individual’s responsibility towards health promoting behavior [49,50].

Health promotion programs must direct resources to areas requiring immediate attention. Moreover, prioritization optimizes the time for solving problems once the need is ranked from the highest to the lowest. Thus, we can point to ALS students having the highest need for improving both lifestyle and quality of life. Pouresmaeil et al. [51] prioritized health-promoting lifestyles in medical science students and found physical health and health responsibility required immediate intervention. Additionally, compliance with WHO/PAHO health promotion guidelines varied across academic disciplines. The HS schools were the only ones that met every guideline [30,31], while the AEH, ET, and SAS schools ranked first in need for increasing health promotion actions. A food menu without healthy choices, no display of health promotion posters, and no availability of a health promotion course in the curriculum were the top areas in need of fixing. The first two are relatively easy to fix compared to the inclusion of a health promotion course that requires professors and university political effort. An international study carried out in 2016 with 141 universities from 48 countries showed the ability and capacity for improving health, the development of health policies, and health promotion research as the ideal areas in need of solutions. Additionally, exercise, healthy diet, mental health, and alcohol abuse prevention were areas in which more work had been done in the past three years [35].

### Limitations of the Study

This study was carried out prior to confinement due to COVID-19 and we were able to survey students from 100% of the university’s schools. However, this is a public university and the results cannot be generalized to students from the private sector. Students from private higher education schools should be included in future studies for analyzing the differences between sectors. Some authors have shown lifestyle and quality of life dissimilarities by semester of study [8,14]. Here, most students were in their second academic year, followed by their first and then third years. A longitudinal study design is recommended for measuring changes over time—even better, during and after the coronavirus pandemic. Finally, it is possible that lifestyle and quality of life were overestimated, as students could have provided socially acceptable answers. However, we consider this bias to be low, since the questionnaire was self-administered, and anonymity was guaranteed at the time of its application.

## 5. Conclusions

Our study adds information on university students’ lifestyle and quality of life, according to and along with the availability of health promotion activities at the schools where they study. Men had healthier lifestyles with more exercise and better stress management; women had better relationships. Exercise and bodily pain were the only better domains in health sciences students. The University should provide sex-sensitive and academic discipline-sensitive actions for improving certain lifestyles and quality of life, who and in what order can now be recognized thanks to the methodology used that allows hierarchization for targeting strategies and resources. Compliance with WHO/PAHO health promotion guidelines varied across disciplines. The University should use HS schools as benchmarking since they achieved the highest outcome. College students are the decision makers of the future and university authorities must guarantee healthy environments and contribute with healthier lifestyles and the best possible quality of life of their students.

## Figures and Tables

**Figure 1 ijerph-17-08024-f001:**
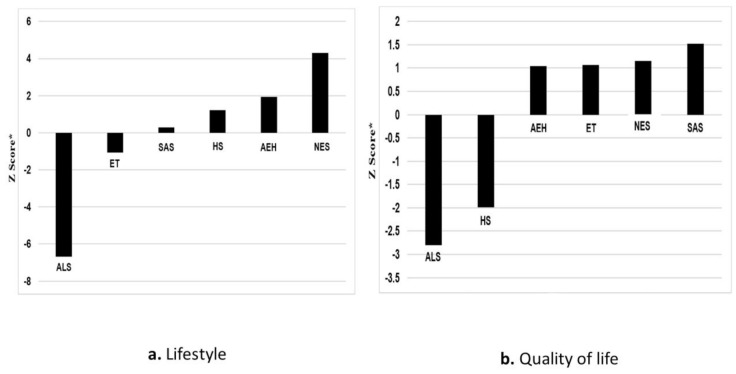
Ranking of students’ lifestyle and quality of life needs by academic discipline (*N* = 5443). * A higher negative Z-score indicates higher priority. AEH, Arts, Education, and Humanities; ALS, Agricultural Sciences; HS, Health Sciences; NES, Natural and Exact Sciences; SAS, Social and Administrative Sciences; ET, Engineering and Technology.

**Figure 2 ijerph-17-08024-f002:**
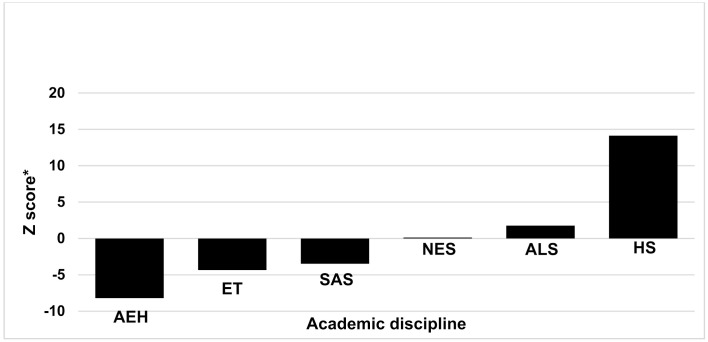
Ranking of schools’ health promotion needs by academic discipline (*N* = 26). * A higher negative Z-score indicates higher priority. AEH, Arts, Education and Humanities; ALS, Agricultural Sciences; HS, Health Sciences; NES, Natural and Exact Sciences; SAS, Social and Administrative Sciences; ET, Engineering and Technology.

**Table 1 ijerph-17-08024-t001:** Lifestyle and Quality of Life in University Students, Stratified by Sex (*N* = 5443).

Domain	Men (*n* = 2186)	Women (*n* = 3218)
Lifestyle (on a 0–100 scale; the higher the score the better)		
Nutrition	49.4 ± 22.5	48.4 ± 23.7
Exercise	45.4 ± 26.7 **	38.5 ± 27.1
Health responsibility	33.8 ± 19.0	33.8 ± 17.9
Stress management	45.6 ± 20.0 **	41.1± 19.2
Interpersonal relationships	67.5 ± 19.5	70.2 ± 19.1 **
Self-actualization	74.5 ± 17.7	73.9 ± 17.7
Quality of life (on a 0–100 scale; the higher the score the better)		
Physical health component	77.4 ± 14.8 **	74. 3 ± 15.3
Physical functioning	91.4 ± 18.2 **	87.1 ± 20.3
Physical role	78.2 ± 36.0	77.2 ± 36.3
Bodily pain	83.7 ± 22.9 *	81.6 ± 24.1
Vitality	59.0 ± 23.5 **	53.9 ± 22.6
Mental health component	68.7 ± 17.8 **	63.7 ± 18.0
Social functioning	79.9 ± 26.4 **	77.0± 27.5
Emotional role	76.8 ± 38.1 **	69.7 ± 41.9
Mental health	67.5 ± 19.8 **	62.1 ± 19.1
General health	56.4 ± 23.2 **	52.6 ± 21.9

* *p* < 0.001; ** *p* < 0.0001.

**Table 2 ijerph-17-08024-t002:** Lifestyle and Quality of Life in University Students, Stratified by Academic Discipline (*N* = 5443).

Academic Discipline						
Domain	AEH (*n* = 440)	ALS (*n* = 390)	HS (*n* = 1360)	NES (*n* = 723)	SAS (*n* = 1220)	ET (*n* = 1177)
Lifestyle (on a 0–100 scale; the higher the score the better)						
Nutrition	50.9 ± 23.7	47.2 ± 21.8	49.5 ± 23.1	50.4 ± 24.8	47.9 ± 23.1	47.6 ± 22.7
Exercise	40.7 ± 27.2	37.6 ± 26.9	42.5 ± 26.9 ^a^	41.6 ± 27.2	41.4 ± 27.4	41.0 ± 27.2
Health responsibility	34.2 ± 19.1	33.5 ± 16.7	33.4 ± 17.6	37.0 ± 20.1 ^b^	33.3 ± 18.5	32.7 ± 17.8
Stress management	42.5 ± 20.7	43.2 ± 18.7	42.6 ± 19.5	44.0 ± 20.1	43.3 ± 19.8	42.3 ± 19.4
Interpersonal relationships	69.4 ± 19.7	67.6 ± 19.9	69 ± 19.3	68.7 ± 19.8	69.1 ± 19.3	69.7 ± 18.7
Self-actualization	74.6 ± 18.8	72.2 ± 17.7	74.6 ± 17.4	73.6 ± 18.6	74.2 ± 17.8	74.4 ± 17.0
Total	54.4 ± 15.7	52.1 ± 14.3	54.0 ± 14.3	55.0 ± 16.0	53.6 ± 15.1	53.6 ± 14.3
Quality of Life (on a 0–100 scale; the higher the score the better)						
Physical health component	75.7 ± 15.3	74.8 ± 14.8	74.8 ± 15.0	75.9 ± 15.0	76.3 ± 15.3	75.6 ± 15.5
Physical functioning	89.3 ± 19.2	88.5 ± 19.5	88.8 ± 19.6	88.4 ± 19.9	89.3 ± 19.2	88.6 ± 20.0
Physical role	75.2 ± 37.6	77.1 ± 36.1	75.6 ± 37.2	79.1 ± 35.8	79.4 ± 34.7	78.1 ± 36.3
Bodily pain	83.1 ± 22.7	82.1 ± 23.4	80.7 ± 24.2 ^c^	82.3 ± 24.7	83.3 ± 23.7	83.7 ± 22.6
Vitality	55.8 ± 23.6	55.4 ± 22.4	55.4 ± 22.7	55.8 ± 24.2	56.1 ± 23.0	56.9 ± 23.1
Mental health component	66.4 ± 18.6	64.1 ± 17.8	64.9 ± 18.0	66.2 ± 18.8	65.9 ± 17.9	66.6 ± 17.9
Social functioning	78.3 ± 28.6	76.2 ± 28.0	77.9 ± 27.1	77.3 ± 26.7	78.4 ± 26.8	79.4 ± 26.5
Emotional role	73.4 ± 40.3	71.9 ± 40.9	71.8 ± 40.9	71.4 ± 41.8	72.2 ± 40.3	74.5 ± 39.6
Mental health	65.7 ± 19.2	61.8 ± 19.2 ^d^	63.2 ± 19.7	65.7 ± 20.0	64.4 ± 19.3	64.9 ± 19.6
General health	55.2 ± 22.1	52.6 ± 22.7	54.1 ± 22.2	55.1 ± 22.8	54.6 ± 22.8	53.2 ± 22.4
Total	70.1 ± 15.1	68.3 ± 14.8	69.0 ± 14.8	70.0 ± 15.2	70.0 ± 14.8	70.2 ± 15.1

^a^ HS > ALS, *p* < 0.02; ^b^ NES > HS, SAS, and ET, *p* < 0.04; ^c^ HS < ET and SAS, *p* < 0.02; ^d^ ALS < ET, NES, and AEH, *p* < 0.05. AEH, Arts, Education, and Humanities; ALS, Agricultural Sciences; HS, Health Sciences; NES, Natural and Exact Sciences; SAS, Social and Administrative Sciences; ET, Engineering and Technology.

**Table 3 ijerph-17-08024-t003:** Health Promotion Guidelines Compliance in Faculties According to Academic Discipline (*N* = 26).

Health Promotion Indicator	AEH (*n* = 4) %	ALS (*n* = 2) %	HS (*n* = 5) %	NES (*n* = 4) %	SAS (*n* = 7) %	ET (*n* = 4) %	Total (*N* = 26) %
1. The school development plan includes environmental health policies	100.0	100.0	100.0	100.0	100.0	75.0	96.2
2. The school offers a health promotion course	0.0	0.0	100.0	25.0	14.3	50.0	34.6
3. The school conducts health promotion research	50.0	50.0	100.0	75.0	14.3	25.0	50.0
4. The school has implemented health promotion programs or campaigns	50.0	50.0	100.0	50.0	28.0	25.0	50.0
5. The school has a health prevention or health care module	25.0	50.0	100.0	50.0	28.6	25.0	50.0
6. The school participates in leadership actions and health advocacy	25.0	100.0	100.0	25.0	42.9	25.0	34.6
7. The school provides a health promotion community service	75.0	50.0	100.0	75.0	71.4	50.0	73.1
8. The school displays no smoking signs	50.0	100.0	80.0	75.0	71.4	100.0	76.9
9. The school displays health promotion posters	0.0	50.0	100.0	0.0	14.3	25.0	30.8
10. The school has areas for sports and exercise	25.0	100.0	100.0	100.0	100.0	100.0	88.5
11. The school cafeteria offers a food menu with healthy choices	25.0	0.0	60.0	25.0	14.3	25.0	26.9
12. The school promotes sports	75.0	100.0	100.0	100.0	100.0	100.0	96.2
Total (mean percentage)	43.8	58.3	95.0	56.3	48.8	50.0	58.7

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
