# Peer review of "Lifestyle, Quality of Life, and Health Promotion Needs in Mexican University Students: Important Differences by Sex and Academic Discipline"

_ijerph, 2020, doi:10.3390/ijerph17218024_

Round 1

Reviewer 1 Report

Thank you for providing this paper to review. I enjoyed reading this paper. The findings are easy to understand, however, need minor revision before getting accepted. My comments/ reviews are as follows:

Introduction:

  1. What kinds of health-promoting lifestyle is prevalent among university students in Mexico? Please provide information about diet preferences, physical activities, screen time, and prevalence of disease caused by unhealthy habits, and add them in the introduction section.
  2. Please also mention the health-promoting activities conducted by government, WHO and PAHO for university students or adolescents in general.

Methods:

  1. I am concerned about the data collection procedures as little information has been provided in the manuscript. Please mention the sampling techniques in the study design section. Was it a convenient sampling of the participants were selected randomly? Also, how much time was allocated to participants to complete this questionnaire? Also please indicate the duration of this study. 
  2. Based on reference 27, I believe the Spanish version of SF-12 might have been used. Is it the same with the PEPS-1 scale? Being a validated questionnaire, I believe no pretest was needed in the study to establish the reliability of the questionnaire. However, please mention about the language version adopted by the questionnaire in the manuscript.
  3. As the study has been conducted in the university students, who are around 20 years, how was the consent taken from the participants? Was it a verbal or written informed consent? 
  4. Please elaborate on information about "a qualified person" in section 2.3 of the manuscript whether it was a researcher or public health officer or specialist.

Discussion:

  1. The sex-stratified table 1 showed low mean score in majority of the variables (lifestyle, QOL, exercise, and stress management) among women. What kinds of health beliefs related to health promotion are perceived by women, barriers contributing to have reduced scores among women and what actions need to be taken to address this issues? Please address these in the discussion section.
  2. The low mean score for health responsibility seems alarming as the participants will turn to adults with time. Moreover, participants from the Health Sciences discipline showed similarly low mean scores as compared to other disciplines. More discussion is needed to address the low health responsibility among the participants. What could be a hindrance or confounding factor for these results? The authors need to clarify them by backing up with other studies.

Author Response

In attached file, reply to reviewer 1 is sent

Thank you

Reviewer 2 Report

Introduction. There is not specified the aim of the research nor hypothesis. Introduction should present the background of the problem you want to research or solve. Why you made this research? What theory is behind ?

Discussion shouldn't be the simple comparison of the result. 

Conclusions in your manuscript are re-written results. 

You interviewed quite large population of students. Write why you wanted to know their lifestyles and quality of life and what is this knowledge for ?

It should be the rigorous academic research. It should provide hypothesis and theory. Finally should demonstrate theoretical and practical implications.

Author Response

In attached file, reply to reviewer 2 is sent

Thank you

Round 2

Reviewer 2 Report

I endorse the manuscript for publication.